# A Comparative Modelling Study of New Robust Packaging Technology 1 mm^2^ VCSEL Packages and Their Mechanical Stress Properties

**DOI:** 10.3390/mi13091513

**Published:** 2022-09-13

**Authors:** Khairul Mohd Arshad, Muhamad Mat Noor, Asrulnizam Abd Manaf, Hiroshi Kawarada, Shaili Falina, Mohd Syamsul

**Affiliations:** 1Institute of Nano Optoelectronics Research and Technology (INOR), Universiti Sains Malaysia, Sains@USM, Bayan Lepas 11900, Pulau Pinang, Malaysia; khairulmohdarshad@student.usm.my; 2Faculty of Mechanical and Automotive Engineering Technology, Universiti Malaysia Pahang, Pekan 26600, Pahang, Malaysia; muhamad@ump.edu.my; 3Collaborative Microelectronic Design Excellence Centre (CEDEC), Universiti Sains Malaysia, Sains@USM, Bayan Lepas 11900, Pulau Pinang, Malaysia; eeasrulnizam@usm.my (A.A.M.); shailifalina@usm.my (S.F.); 4Faculty of Science and Engineering, Waseda University, Shinjuku, Tokyo 169-8555, Japan; kawarada@waseda.jp; 5The Kagami Memorial Laboratory for Materials Science and Technology, Waseda University, 2-8-26 Nishiwaseda, Shinjuku, Tokyo 169-0051, Japan

**Keywords:** mechanical stress, VCSEL, diffuser

## Abstract

Face recognition is one of the most sophisticated disciplines of biometric systems. The use of VCSEL in automotive applications is one of the most recent advances. The existing VCSEL package with a diffuser on top of a lens intended for automotive applications could not satisfy the criteria of the automotive TS16949: 2009 specification because the package was harmed and developed a lens fracture during 100 thermal cycle tests. In order to complete a cycle, the temperature rises from −40 °C to 150 °C and then rises again from 150 °C to 260 °C. The package then needs to be tested 500 times to ensure it fits the requirements without failing in terms of appearance or functionality. To this extent, the goal of this research is to develop packaging for 1 mm^2^ VCSEL chips with a diffuser on top that prevents fractures or damage to the package during heat cycle testing with multiple materials. The package was created using the applications SolidWorks 2017 and AutoCAD Mechanical 2017. The ANSYS Mechanical Structural FEA Analysis program simulated all packages for mechanical stress to guarantee that all packages generated were resilient to high temperature conditions. All packages exhibit no abnormalities and are robust for various temperatures ranging from low to high. Therefore, these packaged 1 mm^2^ VCSEL chips with a diffuser on top provide an effective approach for the application of VCSEL suitable in high temperature conditions.

## 1. Introduction

Vertical-cavity surface-emitting lasers (VCSELs) are high-performance semiconductor devices consisting of different epitaxial layers grown on n-type GaA or InP substrates. VCSEL technology was originally used for data transmission, but a large number of potential applications in other areas have recently been discovered [1]. In contrast to edge-emitting laser diodes, which emit light from the edge of the chip, surface emitters produce light from the surface of the chip, which results in lower production costs, better beam quality, and lower output power. VCSELs are also surface-mountable components that combine LED and laser properties. VCSEL technology is an excellent alternative for applications that require high-speed modulation, such as smartphones, drones, and augmented reality (AR) and virtual reality (VR) devices. VCSELs can be used as a light source for facial recognition in mobile devices, uniformly illuminating the face with infrared light so that the camera can record the user’s important features. The image is then compared with the user’s image recorded in the system; if both of the images match, the device is unlocked. VCSELs combine the best of two lighting technologies: the high-power density and convenient packaging of infrared LEDs (IREDs) and the broad spectrum and speed of lasers [2]. It is the responsibility of the security system to secure all information or data. Biometric systems can be used to limit passenger access to restricted areas, at border control points, or at airports. DNA, iris recognition, face recognition, fingerprint recognition, voice recognition, and signature recognition are some of the biometric technologies that have been introduced and will be continuously implemented in the future [3,4].

Biometric technologies will serve the same purpose for identification and verification. However, the technology and application elements will be distinct from those of others, such as traditional identification, e.g., a passport or identity card (IC). Implementation of VCSEL with packaging with diffuser was an innovation for recognition systems or biometrics with more accuracy compared with devices using infrared such as iris recognition. In previous research, VCSEL technology has been used in the automotive industry, such as for time of flight (ToF). ToF sensors are increasingly being employed for a number of in-cabin vehicle monitoring applications, including tracking the driver’s gaze direction and head position. A ToF device is a sensor that detects light wavelengths. Light waves are modified at high frequencies (in the lower to higher MHz range) and reflected by a target. The sensor receives and processes a portion of the original signal. Because of the time it takes for the signal to travel from the emitter to the object and back to the receiver, the reflected signal is phase-shifted with respect to the modulation signal. The phase shift is proportional to the distance, allowing the distance between the ToF sensor and the object to be calculated [5]. Previously, the devices employed by ToF were LED or infrared. However, LED and infrared have flaws and limits when used in ToF. As a result, the employment of VCSEL is employed to solve the problem. The employment of VCSEL satisfies the requisite criteria due to its coherent and exact properties, as well as its ability to cover the angle required by the ToF [6]. Other than that, VCSELs can be utilized in automotive IR lighting applications that aim to improve driver safety by providing weather independent vision, such as seeing through adverse circumstances (i.e., fog) and reducing blind spots. When compared to standard high beams, such IR lighting can improve vision at a considerably greater distance without interfering with other vehicles [7]. 

The package’s design and dimensions were based on the present packaging for chip size of 0.5 mm^2^ and 0.75 mm^2^. The package was designed based on manufacturing capability using Six Sigma rules [8] and it was also capable of producing the package with high quality and meeting automotive requirements such as ISO TS 16949:2009, where the quality system requirement was jointly developed by the US, German, French, and Italian automotive industries in a concerted effort to improve quality and ensure the integrity of supplies to the industry. The requirement applies to any company that makes components, assemblies, or parts for the automobile sector. The diffuser on top of the chip was deemed a novel concept or idea in comparison to another existing packaging where the diffuser was separate or attached to the lens for other chip dimensions. The major purpose of the current VCSEL package is to comply with the automotive requirement specification, which specifies the use of face and gesture recognition to identify the owner or passenger of their respective vehicles. Our goal is to develop and deliver the most recent innovations in biometric systems, particularly the use of VCSEL packaging with the diffuser on the VCSEL chip having a dimension of 1 mm^2^ that can fulfill the requirements of automotive specifications where the package must qualify for thermal cycle testing between −40 °C and 260 °C for 1000 cycles. 

## 2. Current Challenge on 1 mm^2^ VCSEL Package

The current 1 mm^2^ VCSEL package was developed with the diffuser attached to the lens. The package was made using ceramic, lens, and supportive materials including potting, lens-attached glue, and die-attached glue. The VCSEL package with a diffuser on the lens was subjected to a thermal cycle from −40 °C to 260 °C to fulfill the requirements of automotive reliability. Figure 1a,b shows the structure of the diffuser on the inside lens such as a microstructure lens based on an image from a scanning electron microscope (SEM).

An optical diffuser diffuses light by shuffling its wavefront and reducing its spatial coherence. It acquires random or pseudo-random fluctuations in the optical phase for different parts of a given incoming light profile [9]. When a highly spatially coherent laser beam collides with a diffuser, the light emitted by the diffuser may lose its beam characteristics and move in a wide range of directions. The degree and specific properties of diffusion, on the other hand, might vary significantly among devices [10]. Optical diffusers come in a variety of geometrical forms to suit a variety of purposes. Diffuser plates, for example, are typically round or rectangular in shape with a thin thickness of a few millimeters [11,12] There are also diffuser coatings that may be applied to a variety of surfaces, including metals and plastics [13]. The diffuser was created and developed using epoxy as the lens, which has transparent behavior and a microstructured shape on top of the lens. Figure 2 shows the SEM image of the microstructure of the diffuser on the lens.

The package must undergo and pass the thermal cycle test with −40 °C to 260 °C at 500 h for industrial and 1000 h for automotive needs to meet the criteria of the reliability specification. During this, the package was examined, and it was discovered that the lens had a break towards the beginning of the thermal cycle test, which was less than 100 h. Figure 3a,b depicts a top view of the unit as well as a side view of a cross-section of the unit. The crack was discovered on the package’s top and side and crept down to the unit’s bottom, as shown in Figure 3c, from the A position to the E position.

Based on the thermal cycle test shown, the package did not meet the requirements of the reliability specification for automotive specifications. Thus, new research and ideas have been filed to address all of the issues raised by the current VCSEL package diffuser on the lens. In this investigation, a new concept and packages were presented, where the implementation of microlenses and VCSEL array chips [14] used were the same concept used for biometric systems for face recognition. 

## 3. Theoretical Analysis

Stress can be interpreted as an internal resistance force produced in a component as a result of deformation under the action of loading. Due to this load, an internal resisting force (IRF) is formed, and the body tends to restore its original shape if deformation reaches the elastic limit [15,16]. All units of measurement are in N/m^2^ or Pa or MPa. The mechanical stress in the simulation was presented based on the hook equation, as shown in Equation (1).

(1)
σ=E × ϵ

where *E* is the modulus and *ϵ* the mechanical strain from thermal loading. The package was simulated under mechanical stress during the resistance to soldering heat (RTSH) process. Three different materials were introduced: ceramic, lead frame, and printed circuit board (PCB). All of the packages are identical, and the diffuser was grown or formed on top of the chip rather than the lens as in the prior packaging. Supporting other materials, including epoxy for the lens and adhesive, are classified as conductive or nonconductive. The whole package, including the diffuser, was generated by using SolidWorks 2017 and AutoCAD 2017. The dimensions of all packages were ensured to meet the manufacturing capabilities for semiconductor backend processes such as die-attached (DA), wire- bond (WB), lens-attached, and potting with requirements for measurement within a Six Sigma process specification. According to automotive specifications, the maximum temperature stain for the inner component as a dashboard component is only 200 °C [17]. Therefore, to ensure the robustness of the package, the measurement temperature ranges were chosen to be between −40 °C and 260 °C. 

The diffuser was developed on the top chip for the first time, rather than using the lens as a medium for a diffuser or separate component. This novel design was developed to avoid any cracks or damage during the thermal cycle test, specifically to fulfill the high demands of the automobile sector, which demanded that all components remain undamaged at temperatures as high as 200 °C and as low as −40 °C. The diffuser was created and designed to have a diameter of 0.03 m and a height of 0.01 m. It is shown in Figure 4a,b from a cross-sectional view, to fulfill and be workable as a diffuser on the lens as per the current VCSEL package.

During the RSTH process, the whole VCSEL package was placed in extreme conditions of cold at −40 °C to a normal temperature of 25 °C, then to 150 °C, the standard temperature in oven conditions, and finally to a high temperature of 260 °C. Each package consists of different components that were attached to three types of VCSEL packages, as seen in Figure 5. The design of all packages was created to withstand all loads, notably mechanical stress at high temperatures.

## 4. Material and Experiments

The structural design and mechanical stress modeling, using various forms of packaging for VCSEL packaging utilized the 1 mm^2^ chip and the novel. Table 1 shows the structure and list of materials for each form of packaging, with all materials, including the lens and adhesive being the same for all packages. They are expressed in terms of thermal conductivity (W/m.K), coefficient of linear thermal expansion (CTE), and modulus (MPa); in addition, the behavior and characteristics stated are linear elastic. Thermomechanical Analysis (TMA)^1^ and Dynamic Mechanical Analysis (DMA)^2^ are methods for measuring the properties of materials based on temperature and pressure changes, such as epoxy for die attachments, lens, diffusers, and dielectrics, as illustrated in Figure 6a,b. The diffuser material was employed in the same way as a lens would be, with translucent characteristics. The reflector is solely used for PCB package and is built of black polyphthalamide moulded metallized (PPA) with a thin surface gold coating to prevent light absorption. 

During the simulation procedure, the ANSYS Mechanical Structural FEA Analysis programme was utilized to simulate all package mechanical stress elements. The package was installed on a metal core print circuit board (MCPCB) with a dimension of 20 mm × 20 mm at the start of the simulation procedure, as illustrated in Figure 7. Figure 7 depicts each suggested packaging model installed on the MCPCB board prior to simulation. It is done to guarantee that each packaging model is simulated consistently and correctly, and to avoid a gap in the stress value on the packaging model. The layers of MCPCB vary in thickness from 40 µm to 70 µm to 76 µm, with aluminum at 1500 µm. 

Tin Silver Copper (SAC) is a lead-free alloy with a composition of 96.5% Tin + 3.0% Silver and 0.5% Copper, capable of withstanding a temperature of 260 °C. The procedure employed the package temperature at 150 °C (based on the process curing temperature for die-Attached, lens-Attached, potting, and so forth), and thermal loading was a homogeneous temperature application for the whole package (150 °C to 260 °C and 150 °C to −40 °C). The mechanical stress simulation methodology does not use cyclic simulation because material properties are linear, and the stress here represents the condition when the heating up from 150 °C to 260 °C at the maximum temperature for RTSH. The measured area is the die-attached surface, chip, and lens, which represents the full package structure for the identification of fractures or damage, as well as being able to deliver accurate results throughout the simulation process depending on the section or area indicated.

## 5. Results

There is no substantial difference between all the packages with a wide range of temperatures. At higher temperatures, the range of values for stress on all packages is 0 to 19.023 MPa, whereas at lower temperatures, it is 0 to 318.99 MPa. The stress values are uniform and homogenous throughout. Figure 8a,b depicts the mechanical stress simulation for die-attached (DA) in principal stress during the RTSH at 260 °C and −40 °C, respectively, and the graph in Figure 9 depicts the stress value for the package versus temperature.

The chip, which is considered the middle of the package, was the next area that was assessed. Similarly, the die-attached surface, chip, and lens were assessed for the principal stress for each type of package. At RSTH, none of these stress value differences were significant for all packages, as the range of stress values started from 0 to 31.673 MPa at higher temperatures, and at lower temperatures, the value of stress for all packages started from 0 to 211.42 MPa. In contrast to lead frame and PCB, ceramic packages have lower stress values of 2.567 MPa at higher temperatures. Figure 10a,b depicts the stress value at higher and lower temperatures, respectively. The stress values vs. temperatures for all packages are shown in Figure 11.

The lens is the final component to be analyzed, and it is the most crucial factor in evaluating the package’s durability since it is prone to breaking and destruction in the case of severe temperature and pressure changes. This will have a severe influence on the package’s durability. According to the simulation findings obtained on the lens area, the stress values obtained at high and low temperatures are uniform and do not indicate any aberrant stress results on the lens region. At higher temperatures, it ranges from 0 to 1.6045 MPa, whereas at lower temperatures, it ranges from 0 to 128.58 MPa for all packages. Figure 12a,b depicts the stress values for all packages at high and low temperatures, respectively. The stress values versus temperatures are shown in Figure 13. 

According to the findings, all packages produced the same values and were standardized at each stage of the simulation. This proves that the proposed novel packaging design with the placement of the diffuser on the top of the chip prevents lens fractures during the heat cycles. The microstructure of the diffuser controls the efficacy of the optical qualities employed in biometric systems, but it is also a part that is easily broken and damaged if it is based on a weak-based material such as epoxy or glass. Therefore, the transfer of the microstructure on the diffuser from the lens to the top of the chip is beneficial because the chip has a strong base and is able to withstand significant changes in temperature and pressure. Overall, all of the proposed models produce uniform pressure values on each area (die-attached, chip, and lens) without a critical point that permits fractures to form throughout the heat cycles. The stress value of PCB is the lowest when compared to ceramic and lead frame because the CTE value on FR4 PCB is 1.20 × 10^−^^5^ whereas ceramic is 1.80 × 10^−^^5^ and lead frame is 2.35 × 10^−^^5^, as indicated in Figure 12. Significant stress value differences can be seen for die-attached (Figure 9) and chip (Figure 11), however not in Figure 13. This is due to the identical CTE employed in all packaging variants. As a result, the values obtained have a negligible difference. Nevertheless, all of these packaging types are capable of exerting maximal pressure impact at both low and high temperatures. Ultimately, the package passed the automotive specification thermal cycle test and differed from the present VCSEL for 1 mm^2^ with the diffuser package.

## 6. Conclusions

The primary purpose of this research is to discover VCSEL packages with a functioning diffuser that can endure high temperatures and pressure variations, allowing them to fulfill the automotive specifications outlined in the TS16949:2009 standard. This study looked at three different types of packaging models proposed for different materials: ceramic, lead frame, and PCB, all of which are capable of maintaining and providing equal stress values for each stage or area during stress simulation and providing the best impact on the entire package, particularly on the part or area with the microstructure, which is the chip. According to the model provided in this project, the pressure value given in all models, including low and high temperature, is the same in each packing model. Furthermore, there are no critical points in this packaging model that might induce cracking during the thermal cycle process, demonstrating that the suggested packaging model can withstand any temperature and condition. Identifying parts of the package that are prone to cracking and destruction is one of the major procedures necessary before the design and manufacturing processes are executed. It can save time, energy, and money throughout the packing process. As a result, mechanical stress simulation is a wise step that can discover areas or sections that require maintenance if there are fractures and damage. Furthermore, it may assess if the material used in the packing process is precise and can satisfy the standards, particularly if it can handle high temperatures and pressures. The three packages were designed to satisfy the demand for a thermal cycle that could accept heat and stress while also preventing lens cracking or damage.

## Figures and Tables

**Figure 1 micromachines-13-01513-f001:**
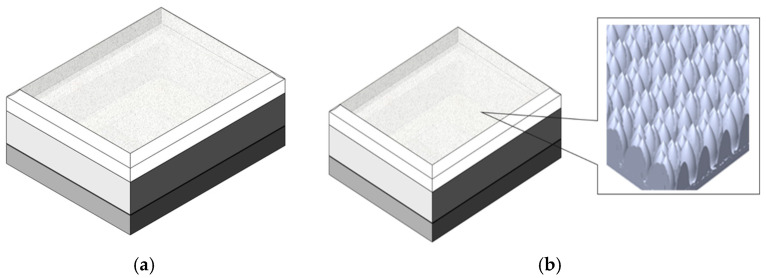
(**a**) The illustration of VCSEL package; (**b**) the illustration of microstructure of lens on top of VCSEL package.

**Figure 2 micromachines-13-01513-f002:**
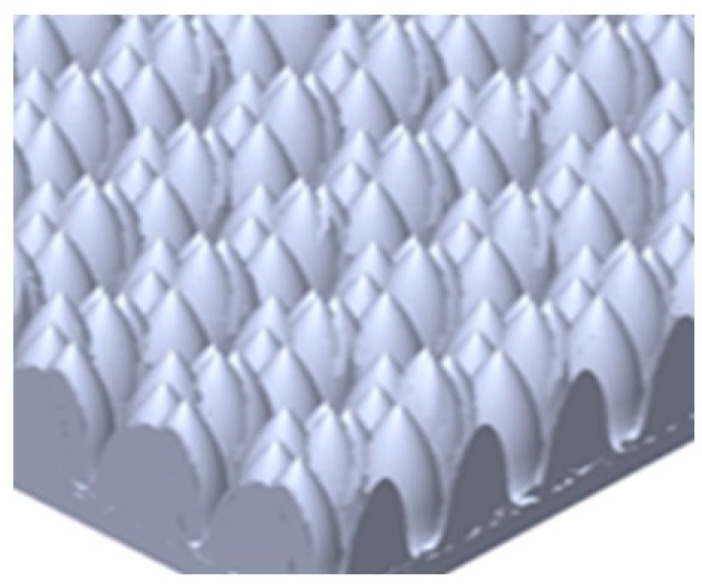
The illustration of the microstructure of the lens on top of the VCSEL package.

**Figure 3 micromachines-13-01513-f003:**
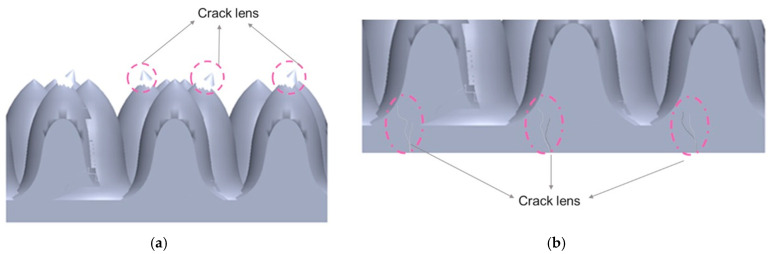
(**a**,**b**) The illustration of microstructure has crack and damage of microstructure lens.

**Figure 4 micromachines-13-01513-f004:**
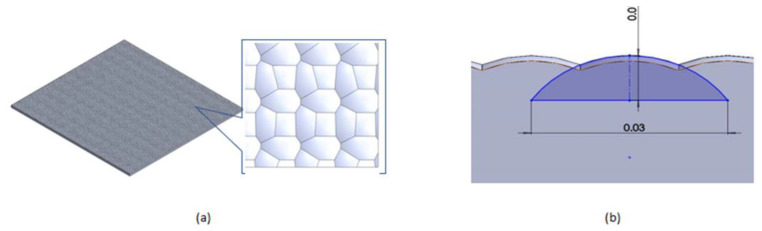
The convex share of the microstructure of diffuser on top lens. (**a**) views from the top of the chip with microstructure diffuser and (**b**) the cross-sectional view of diffuser microstructure.

**Figure 5 micromachines-13-01513-f005:**
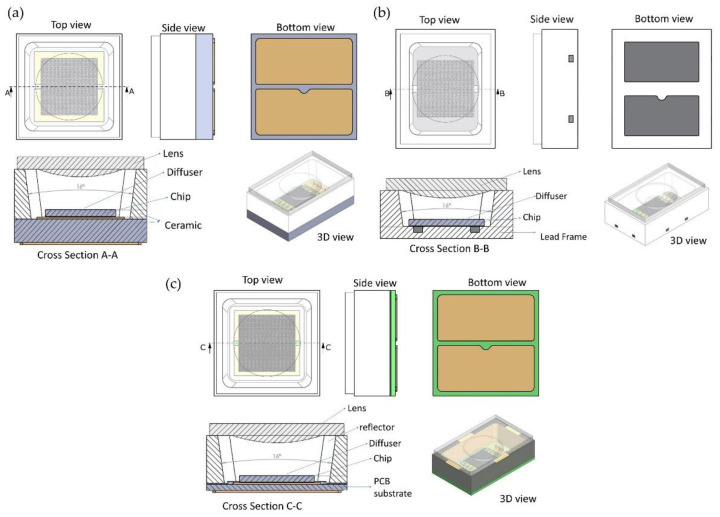
Three types of VCSEL packages are proposed, which incorporate all of the key components attached to the package, including the diffuser on top of the chip: (**a**) ceramic, (**b**) lead frame, and (**c**) PCB.

**Figure 6 micromachines-13-01513-f006:**
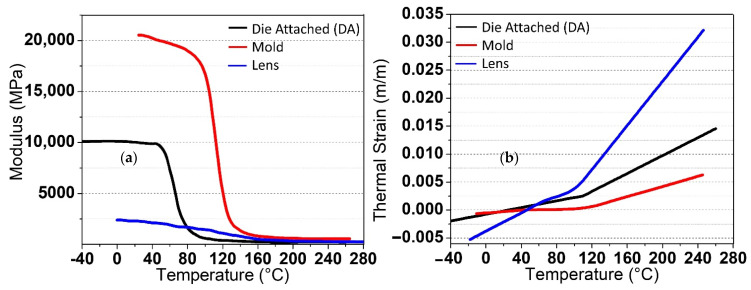
Material characteristics for the DA, mold, and lens using (**a**) Dynamic Mechanical Analysis (DMA) and (**b**) Thermomechanical Analysis (TMA).

**Figure 7 micromachines-13-01513-f007:**
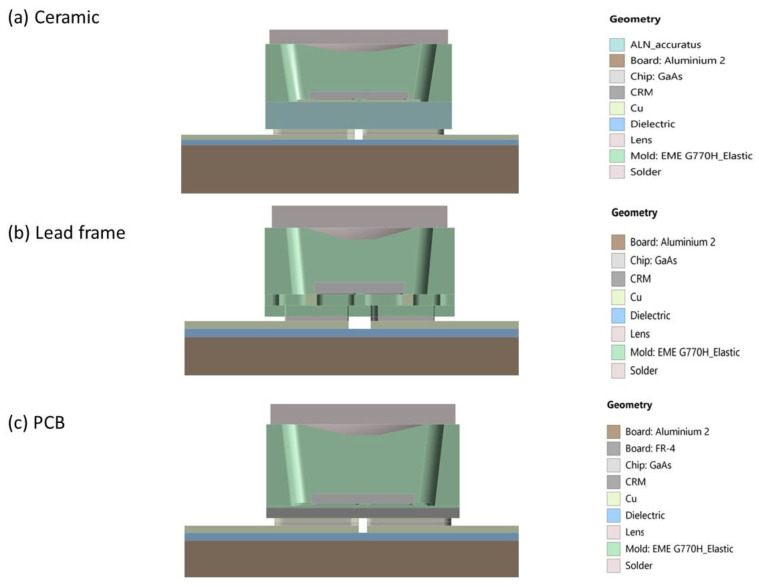
Cross section illustration of the three types of VCSEL packages (**a**) ceramic, (**b**) lead frame and (**c**) PCB were mounted on top of an MCPCB with a dimension 20 mm × 20 mm. The specifications of the material are illustrated on the right side for ceramic, lead frame, and PCB respectively.

**Figure 8 micromachines-13-01513-f008:**
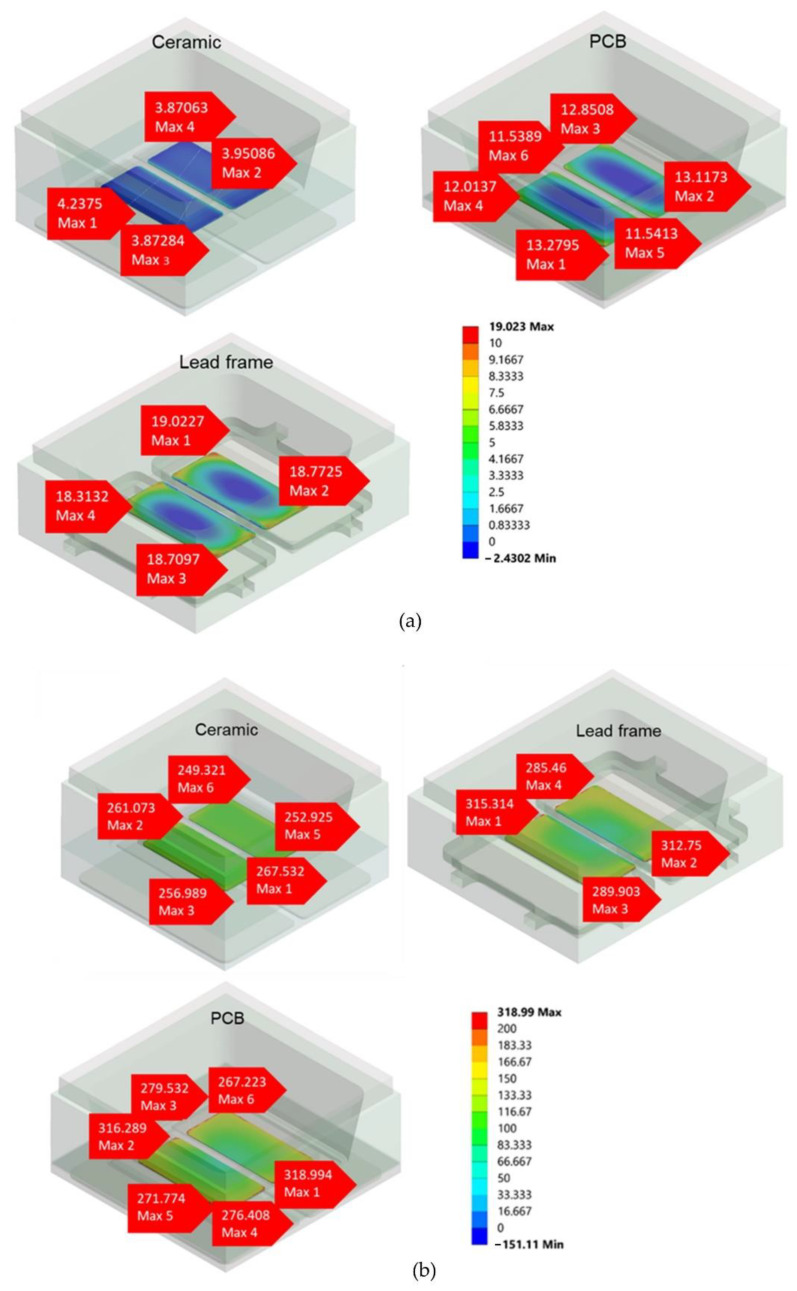
The 3D image of the stress value at the die-attached (DA) area for three types of VCSEL packages during mechanical stress simulation at (**a**) high temperature of 260 °C and (**b**) low temperature of −40 °C.

**Figure 9 micromachines-13-01513-f009:**
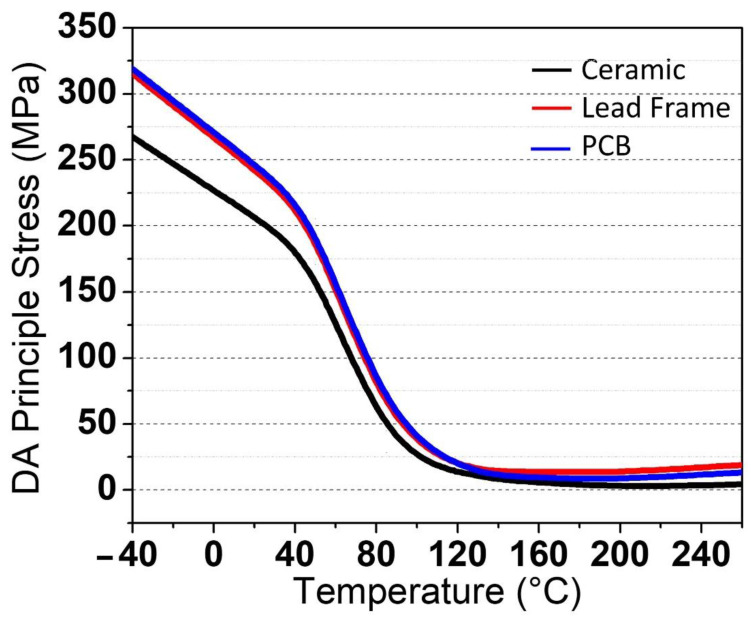
Die-attached (DA) stress values versus temperatures ranging from −40 °C to 260 °C during the mechanical stress simulation.

**Figure 10 micromachines-13-01513-f010:**
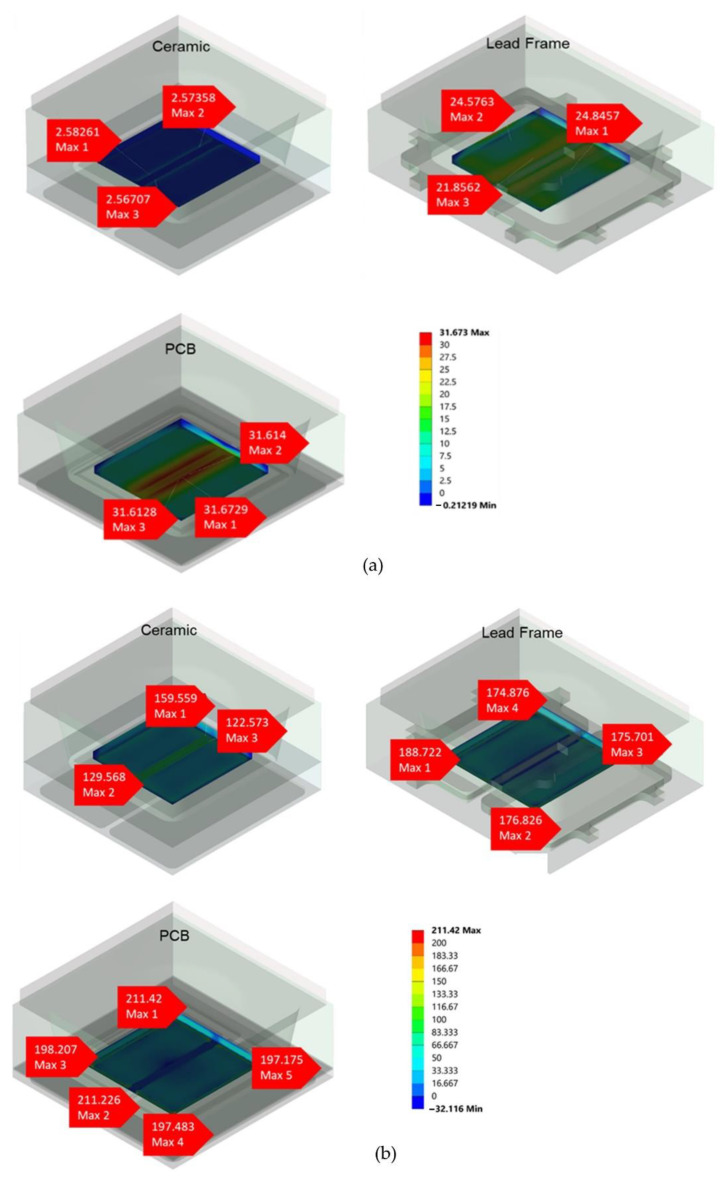
The 3D image of the stress value at the chip surface area for three types of VCSEL packages during mechanical stress simulation at (**a**) high temperature of 260 °C and (**b**) low temperature of −40 °C.

**Figure 11 micromachines-13-01513-f011:**
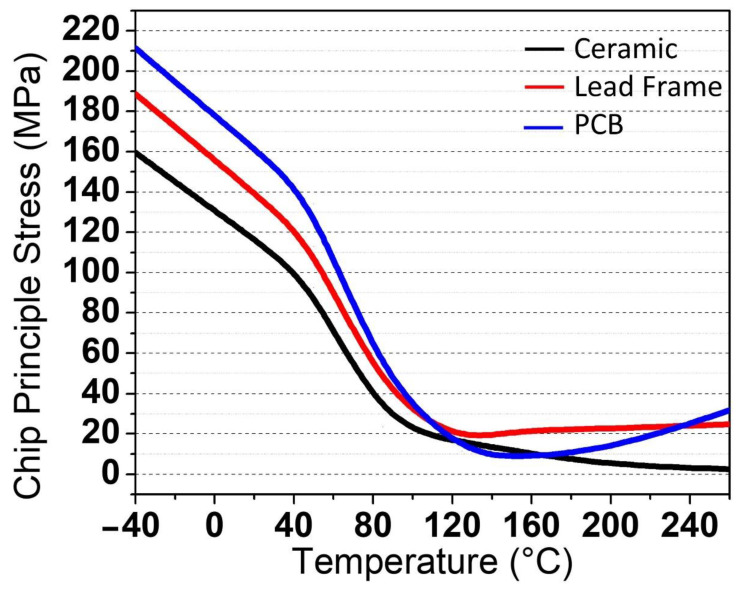
Chip surface stress values versus temperatures ranging from −40 °C to 260 °C during the mechanical stress simulation.

**Figure 12 micromachines-13-01513-f012:**
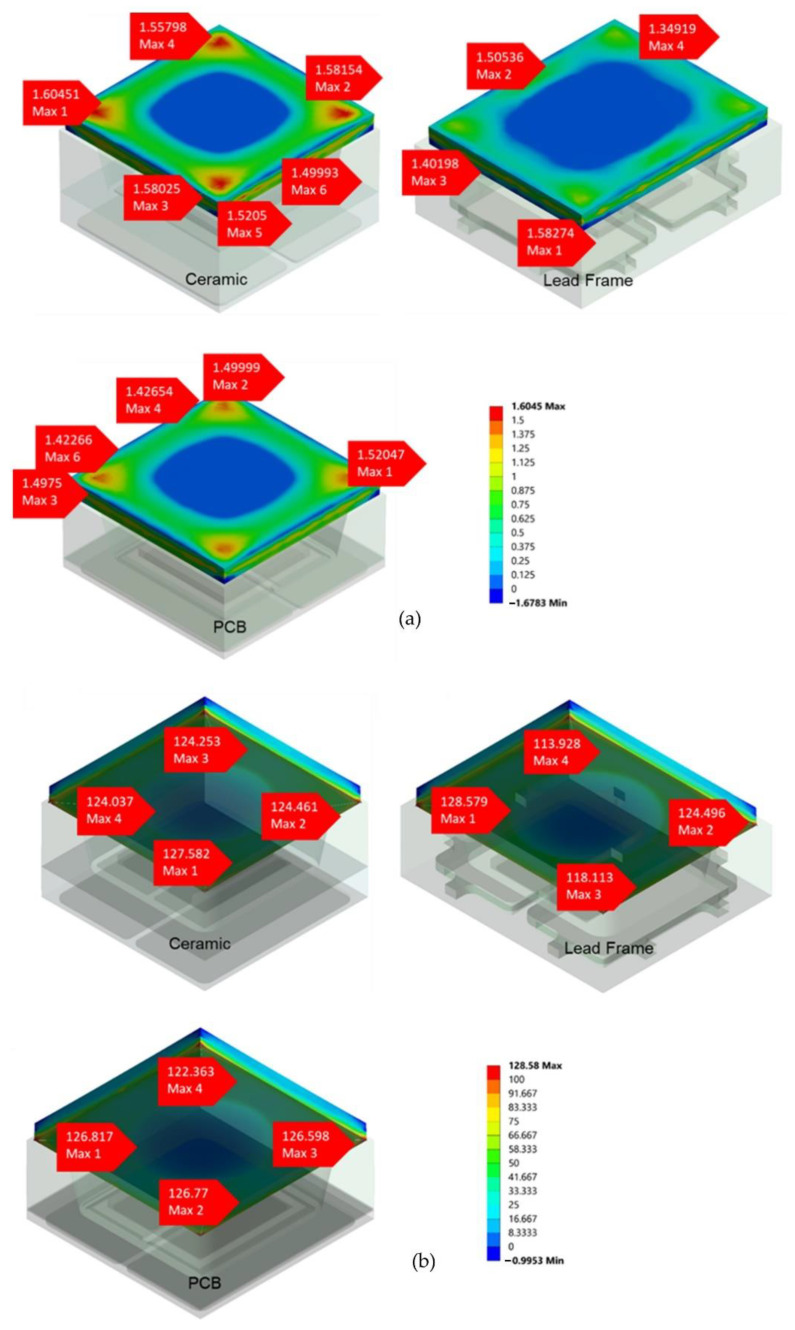
The 3D image of the stress value at the lens area for three types of VCSEL packages during mechanical stress simulation at (**a**) high temperature of 260 °C and (**b**) low temperature of −40 °C.

**Figure 13 micromachines-13-01513-f013:**
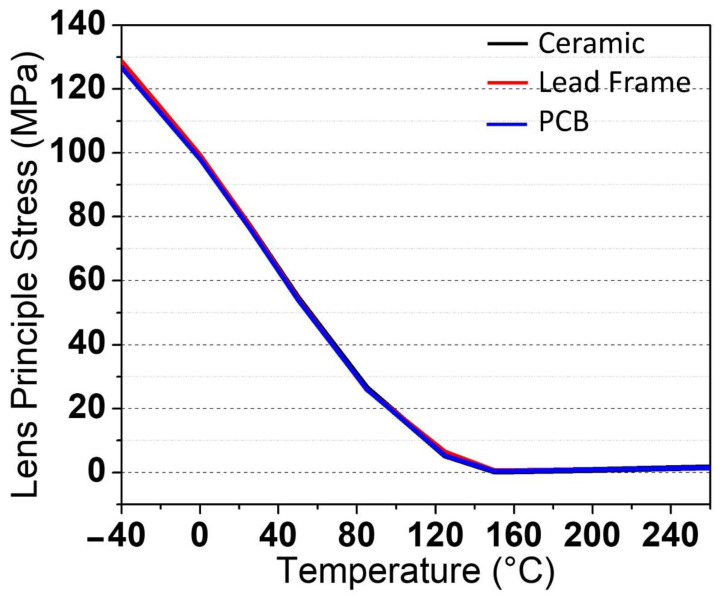
Stress values of lens versus temperatures ranging from −40 °C to 260 °C during the mechanical stress simulation.

**Table 1 micromachines-13-01513-t001:** Material selection for three types of VCSEL packaging was created based on features and characteristics parameters for thermal conductivity (W/m.k), CTE, and modulus.

Material	Thermal Conductivity (W/m.K)	CTE (C-1)	Modulus (MPa)
Chip: GaAs	55.00	5.73 × 10^−6^	85.90
Die-attached (DA) epoxy conductive	1.67	TMA	DMA
Metallization: Cu	380.00	1.80 × 10^−^^5^	110.00
Substrate: AIN	140.00	4.50 × 10^−6^	330.00
Mold:	0.90	Thermomechanical Analysis, TMA ^1^	Dynamic Mechanical Analysis, DMA ^2^
Lens: epoxy	0.90	Thermomechanical Analysis, TMA ^1^	Dynamic Mechanical Analysis, DMA ^2^
Diffuser	0.90	Thermomechanical Analysis, TMA ^1^	Dynamic Mechanical Analysis, DMA ^2^
Solder: lead free	56.00	2.10 × 10^−5^	40.75
Dielectric	1.30	4.00 × 10^−5^	Dynamic Mechanical Analysis, DMA ^2^
Board: Fr4	2.70	1.20 × 10^−^^5^	18.60
Board: Aluminum	205.00	2.35 × 10^−^^5^	70.60

## Data Availability

Not applicable.

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
