# Peer review of "A Comparative Modelling Study of New Robust Packaging Technology 1 mm2 VCSEL Packages and Their Mechanical Stress Properties"

_micromachines, 2022, doi:10.3390/mi13091513_

Round 1

Reviewer 1 Report

This paper describes an effort to use FEM modeling to develop a robust VCSEL packaging technology that meets the industry standards for use in automotive applications.  The following comments are offered to improve the current draft of the paper.

1.  Title:  The title can be improved by being more specific to the study.  The study does not contain experimental results from the new package designs that verify the findings from the FEM study, so the title should reflect that the study is a modeling-only study.  As currently written, the title implies that a new package design was fully developed and verified experimentally.  The current draft is weak in making the connection between VCSEL technologies and automotive applications.

2.  The first sentence of the introduction is probably not necessary.  As such, references 1 and 2 are also probably not necessary.  One could also argue that references 3 and 4 are probably not necessary.  The introduction would be much stronger if it contained references that show how VCSEL technologies are being used in optics-based biometric applications.

3.  The images in Figure 3 are from a failed VCSEL package, which provides the technical rationale for the study.  However, the paper does not indicate that prevalence of this mode of package failure.  As such, it is not possible to determine if the proposed approach addresses the most significant modes of package failure.

4.  The units of modulus are MPa not Mpa.  And the authors should check for the proper way to write the name of the FEM modeling software used in this study (ANSYS vs Ansys).

5.  Figure 7 lacks the resolution to be useful.  All three images in Figure 7 look essentially the same.

6.  The last part of the following sentence that defines cleanroom classifications with respect to particulate concentration is not necessary:  "The values were calculated using the standard cleanroom requirement for the industrial manufacturing backend process, which is 10,000 particles per cubic meter (ISO 7). This means that no more than 10,000 particles bigger than 0.5 μm are  permitted in a 1 ft3 container, and no more than 352 000 particles are allowed in a 1 m3  contained." 

7.  The authors should refrain from using the words "made" and "measured" in reference to the designs described in this manuscript.  Since the three designs described in this paper were not fabricated into testable prototypes, the structures described in this paper were not "made" but rather configured into structures for modeling purposes.  Likewise, the authors did not make "measurements" but rather used simulations to predict the behavior of their model structures.  Use of the words "made" and "measure" risk confusing the reader into thinking that the data presented in Figures 9, 11 and 13 are from actual measurements on fabricated devices.

Author Response

Response to Reviewer 1 Comments

This paper describes an effort to use FEM modeling to develop a robust VCSEL packaging technology that meets the industry standards for use in automotive applications.  The following comments are offered to improve the current draft of the paper.

Our response: We would like to thank the reviewer for taking the time and making the effort to evaluate this manuscript. For the reviewers' convenience, we have attached the responses to all of the insightful comments and observations. We would also like to express our sincere regret for any shortcomings on our part.

  1. Title: (a) The title can be improved by being more specific to the study. The study does not contain experimental results from the new package designs that verify the findings from the FEM study, so the title should reflect that the study is a modeling-only study. (b)  As currently written, the title implies that a new package design was fully developed and verified experimentally.  The current draft is weak in making the connection between VCSEL technologies and automotive applications.

Response (A) :

Our response: We would like to thank the reviewer for taking the time and making the effort to evaluate this manuscript. Here our response amended on Title of manuscripts.

A Comparative Modelling Study of New Robust Packaging Technology 1mm2 VCSEL Packages and Their Mechanical Stress Properties.

Response (b) The current draft is weak in making the connection between VCSEL technologies and automotive applications.

Our response: We would like to thank the reviewer for taking the time and making the effort to evaluate this manuscript. Here our response was adding in at introduction stage.

In previous research, VCSEL technology has been used in the automotive industry such as time of flight (TOF). ToF sensors are increasingly being employed for a number of in-cabin vehicle monitoring applications, including tracking the driver's gaze direction and head position. A ToF device is a sensor that detects light wavelengths. Light waves are modified at high frequencies (in the lower to higher MHz range) and reflected by a target. The sensor receives and processes a portion of the original signal. Because of the time it takes for the signal to travel from the emitter to the object and back to the receiver, the reflected signal is phase-shifted with respect to the modulation signal. The phase shift is proportional to the distance, allowing the distance between the TOF sensor and the object to be calculated [ ]. Previously, the gadget employed by TOF was LED or infrared. However, LED and infrared have flaws and limits when used in TOF. As a result, the employment of VCSEL is employed to solve the problem. The employment of VCSEL satisfies the requisite criteria due to its coherent and exact properties, as well as its ability to cover the angle required by the TOF [ ]. Other than that, VCSELs can be utilized in automotive IR lighting applications that aim to improve driver safety by providing weather independent vision, such as seeing through adverse circumstances (i.e. fog) and reducing blind spots. When compared to standard high beams, such IR lighting can improve vision at a considerably greater dis-tance while without interfering with other vehicles [ ]

  1. The first sentence of the introduction is probably not necessary. As such, references 1 and 2 are also probably not necessary.  One could also argue that references 3 and 4 are probably not necessary.  The introduction would be much stronger if it contained references that show how VCSEL technologies are being used in optics-based biometric applications.

Our response: We would like to thank the reviewer for taking the time and making the effort to evaluate this manuscript. Here our response was adding in at introduction stage.

Vertical-cavity surface-emitting lasers (VCSELs) are high-performance semiconductor devices consisting of different epitaxial layers grown on n-type GaA or InP sub-strates. VCSEL technology was originally used for data transmission, but a large number of potential applications in other areas have recently been discovered [ 1]. In contrast to edge-emitting laser diodes, which emit light from the edge of the chip, surface emitters produce light from the surface of the chip, which results in lower production costs, better beam quality and lower output power. VCSELs are also surface-mountable components that combine LED and laser properties. VCSEL technology is an excellent alternative for applications that require high-speed modulation, such as smartphones, drones and augmented reality/virtual reality (AR/VR) devices. VCSELs can be used as a light source for facial recognition in mobile devices, uniformly illuminating the face with infrared light so that the camera can record the user's important features. The image is then compared with the user's image recorded in the system; if both matches, the device is unlocked. VCSELs combine the best of two lighting technologies: the high-power density and convenient packaging of infrared LEDs (IREDs) and the broad spectrum and speed of lasers [2 ].

  1. The images in Figure 3 are from a failed VCSEL package, which provides the technical rationale for the study. However, the paper does not indicate that prevalence of this mode of package failure.  As such, it is not possible to determine if the proposed approach addresses the most significant modes of package failure.

Our response: We would like to thank the reviewer for taking the time and making the effort to evaluate this manuscript. Here our response was adding at conclusion.

 According to the model provided in this project, the pressure value given in all models, including low and high temperature, is the same in each packing model. Furthermore, there are no critical points in this packaging model that might induce cracking during the thermal cycle process, demonstrating that the suggested packaging model can withstand any temperature and condition.

  1. The units of modulus are MPa not Mpa. And the authors should check for the proper way to write the name of the FEM modeling software used in this study (ANSYS vs Ansys).

Our response: Thank you for your kind advice. Amended as requested throughout the manuscript.

Location of Change: Throughout the manuscript

  1. Figure 7 lacks the resolution to be useful. All three images in Figure 7 look essentially the same.

Our response: We would like to thank the reviewer for taking the time and making the effort to evaluate this manuscript. Here our response was adding in.

figure 7 depicts each suggested packaging model installed on the MCPCB board prior to simulation. It is done to guarantee that each packaging model is simulated consistently and correctly, and to avoid a gap in the stress value on the packaging model.

  1. The last part of the following sentence that defines cleanroom classifications with respect to particulate concentration is not necessary: "The values were calculated using the standard cleanroom requirement for the industrial manufacturing backend process, which is 10,000 particles per cubic meter (ISO 7). This means that no more than 10,000 particles bigger than 0.5 μm are permitted in a 1 ft3 container, and no more than 352 000 particles are allowed in a 1 m3  contained."

Our response: Thank you for your kind advice. Amended as requested throughout the manuscript.

Location of Change:  remove from manuscript as recommended by reviewer.

  1. The authors should refrain from using the words "made" and "measured" in reference to the designs described in this manuscript. Since the three designs described in this paper were not fabricated into testable prototypes, the structures described in this paper were not "made" but rather configured into structures for modeling purposes.  Likewise, the authors did not make "measurements" but rather used simulations to predict the behavior of their model structures.  Use of the words "made" and "measure" risk confusing the reader into thinking that the data presented in Figures 9, 11 and 13 are from actual measurements on fabricated devices.

Our response: Thank you for your kind advice. Amended as requested throughout the manuscript.

Location of Change: Throughout the manuscript

Reviewer 2 Report

Some suggestions and questions to authors,

1. The full name of VCSEL or acronyms needs to be added to the revised manuscript.

2. Detailed information on materials used in this study should be added to the revised manuscript. For example, the compositions of solder or ceramic should be described.

3. The PCB should have the largest CTE. However, the lowest stress values are found in Fig. 13. Why?

4. Meanwhile, the value of CTE in the lead-frame should be less than that in Ceramic or PCB.  Thus, the stress of lead-frame should have the lowest value, right? However, Fig. 9, 11 and 13 show different results. Any comments?

Author Response

Response to Reviewer 2 Comments

  1. The full name of VCSEL or acronyms needs to be added to the revised manuscript.

Our response: Thank you for your kind advice. Amended as requested throughout the manuscript.

Location of Change: Throughout the manuscript

  1. Detailed information on materials used in this study should be added to the revised manuscript. For example, the compositions of solder or ceramic should be described.

Our response: Thank you for your kind advice. Here our response as requested:

The solder paste SAC (Tin Silver Copper) in the Ratio 96.5% Tin + 3.0% Silver and 0.5% Copper used in this simulation is lead-free and capable of withstanding a temperature of 260 ° C.

  1. The PCB should have the largest CTE. However, the lowest stress values are found in Fig. 13. Why?

Our response: We would like to thank the reviewer for taking the time and making the effort to evaluate this manuscript. Here our response:

Overall, all of the proposed models produce an even picture or pressure value on each portion (die attached, chip attached, and lens attached) without a critical point that permits fractures to form throughout the heat cycle. The stress value of PCB is the lowest when compared to ceramic and lead frame because the CTE value on FR4 PCB is 1.20x10-5 compared to ceramic 1.80x10-5 and lead frame 2.35x10-5 as indicated in table 1 for attributes utilised in this project. Significant changes in figures 9 and 11, but not in figure 13, due to pressure readings measured on the lens surface and the identical CTE employed in all packing variants. As a result, the pressure value obtained has a negligible change, as illustrated in fig 13. However, all of these packaging types are capable of exerting maximal pressure impact at both low and high temperatures.

  1. Meanwhile, the value of CTE in the lead-frame should be less than that in Ceramic or PCB.  Thus, the stress of lead-frame should have the lowest value, right? However, Fig. 9, 11 and 13 show different results. Any comments?

Our response: We would like to thank the reviewer for taking the time and making the effort to evaluate this manuscript. Here our response:

Overall, all of the proposed models produce an even picture or pressure value on each portion (die attached, chip attached, and lens attached) without a critical point that permits fractures to form throughout the heat cycle. The stress value of PCB is the lowest when compared to ceramic and lead frame because the CTE value on FR4 PCB is 1.20x10-5 compared to ceramic 1.80x10-5 and lead frame 2.35x10-5 as indicated in table 1 for attributes utilised in this project. Significant changes in figures 9 and 11, but not in figure 13, due to pressure readings measured on the lens surface and the identical CTE employed in all packing variants. As a result, the pressure value obtained has a negligible change, as illustrated in fig 13. However, all of these packaging types are capable of exerting maximal pressure impact at both low and high temperatures.

Round 2

Reviewer 1 Report

The authors addressed all issues raised in my first review except for the issue associated with Figure 7.  All 3 images in Figure 7 look identical even though the figure legends indicate that there are differences.  The authors should consider including cross sectional images made along strategically located cuts to as to illustrate the materials described in each legend.  In its current form, the utility of Figure 7 is significantly limited.

The authors should refrain from using the word "gadget" as it is a non-technical word.

Author Response

Response to Reviewer 1 Comments

  1. The authors addressed all issues raised in my first review except for the issue associated with Figure 7. All 3 images in Figure 7 look identical even though the figure legends indicate that there are differences.  The authors should consider including cross sectional images made along strategically located cuts to as to illustrate the materials described in each legend.  In its current form, the utility of Figure 7 is significantly limited.

Our response: We would like to thank the reviewer for taking the time and making the effort to evaluate this manuscript. Here is our response to adding in the drawing of Figure 7 with a cross section view on the mounted MCPCB board 20mm x 20mm; (as in docx/manuscript)

  1. The authors should refrain from using the word "gadget" as it is a non-technical word.

Our response: Thank you for your kind advice. Amended as requested throughout the manuscript.

Location of Change: Throughout the manuscript

Reviewer 2 Report

No more comments.

Author Response

Response to Reviewer 2 Comments

(x) Extensive editing of English language and style required

Our response: Thank you for your kind advice. Amended as requested throughout the manuscript.

Location of Change: Throughout the manuscript